# High Prevalence of Myocardial Bridging Detected in an Indonesian Population Using Multi-Detector Computed Tomography

**DOI:** 10.3390/medicina60050794

**Published:** 2024-05-10

**Authors:** Antonia Anna Lukito, Rusli Muljadi, Mira Yuniarti, Nyoman Aditya Sindunata, Andreyano Sarikie, Teodorus Alfons Pratama, Reynaldy Santosa Thio, Jessica Christanti, Gilbert Sterling Octavius

**Affiliations:** 1Interventional Radiology Division, Department of Radiology, Faculty of Universitas Pelita Harapan, Tangerang 15811, Indonesia; 2Department of Radiology, Siloam Hospital Lippo Village, Tangerang 15811, Indonesia; 3Department of Cardiology and Vascular Medicine, Siloam Hospital Lippo Village, Tangerang 15810, Indonesia; 4Thoracic and Cardiovascular Imaging Division, Department of Radiology, Faculty of Universitas Pelita Harapan, Tangerang 15811, Indonesia; 5Department of Radiology, Faculty of Universitas Pelita Harapan, Tangerang 15811, Indonesia

**Keywords:** myocardial bridging, prevalence, multi-detector computed tomography, coronary artery stenosis, cardiac-gated imaging techniques

## Abstract

*Background and Objectives*: Myocardial bridging (MB) is still not yet considered a significant finding in Indonesia both radiographically and clinically. Hence, this article aims to assess the prevalence of MB using multi-detector computed tomography (MDCT) and look at factors contributing to stenosis amongst patients with MB. *Materials and Methods:* This study is cross-sectional in a single centre, with consecutive sampling, looking at all patients who underwent a multi-detector computed tomography (MDCT) scan from February 2021 until February 2023. GraphPad Prism version 9.0.0 for Windows (GraphPad Software, Boston, MA, USA) was used to analyse the results. *Results*: There are 1029 patients with an MB, yielding a prevalence of 44.3% (95%CI 42.3–46.4). The left anterior descending vessel is the most commonly implicated, with 99.6%. Among those with stenosis, the middle portion of the bridging vessel is the most common site of stenosis (*n* = 269), followed by the proximal portion (*n* = 237). The severity of stenosis is more often moderate, with 30–50% (*n* = 238). Females (odds ratio [OR] of 1.8, 95%CI 1.4–2.3; *p*-value < 0.0001), older age (*t*-value 5.6, *p*-value < 0.0001), symptomatic patients (OR 1.4, 95% CI 1.1–1.9; *p*-value = 0.013), and higher mean coronary artery calcium score (*t*-value 11.3, *p*-value < 0.0001) are more likely to have stenosis. The degree of stenosis is significantly higher in the proximal stenosis group than in the middle stenosis group (*t*-value 27, *p*-value < 0.0001). *Conclusions*: Our research demonstrates that MB may prevent atheromatosis of the coronary segment distal to the MB and predispose the development of atherosclerosis in the section proximal to the bridge.

## 1. Introduction

Coronary arteries typically follow a path within the subepicardial layer, but when they traverse intramurally, they are termed myocardial bridges (MBs) [1]. The intramural coronary artery is known as the “tunnelled artery,” and the section of the myocardium that covers the artery is termed the “myocardial bridge” [2]. The exact occurrence of this condition varies depending on the diagnostic method used for MB, with rates of 4.3% with conventional angiography, 21.7% using coronary computed tomography angiography (CCTA), and 44.6% with autopsy or cadaveric dissection [3]. Another meta-analysis of 120 studies found the overall prevalence to be 19% (confidence interval [CI]: 17–21%). Autopsy investigations indicated a prevalence of 42% (CI: 30–55%), CT studies showed 22% (CI: 18–25%), and coronary angiography indicated a prevalence of 6% (CI: 5–8%) [4].

The precise significance of MB remains elusive, as it represents the most prevalent congenital coronary anomaly [5]. In November 2023, the American Heart Association (AHA) introduced a new comprehensive cardiovascular risk-assessment tool. This innovative model integrates various cardiovascular–kidney–metabolic (CKM) health factors, such as age, gender, blood pressure, cholesterol levels, medication usage, diabetes, tobacco consumption, and kidney function [6]. Despite AHA’s emphasis on modifiable risk factors, the absence of any reference to MB in the document leaves the direct link between MB and major adverse cardiovascular events (MACE) uncertain [3,7,8,9].

Myocardial bridging (MB) remains unrecognised as a significant discovery in Indonesia, both in radiographic and clinical contexts. This lack of awareness results in MB being underreported and leads to inconsistent radiographic documentation. A cursory search on PubMed using the keywords “Myocardial bridging” and “Indonesia” yields only six manuscripts, of which only two are pertinent, with one being a case report [10,11]. The scarcity of publications concerning MB in Indonesia reflects the dearth of research and clinical attention in this domain.

Therefore, this article endeavours to evaluate the prevalence of MB through multi-detector computed tomography (MDCT) in a sample drawn from a single-centre study in Indonesia. Additionally, it aims to explore the factors contributing to stenosis among individuals with MB. By doing so, this article aims to enhance the understanding of MB prevalence in Indonesia and to foster awareness of this condition among clinicians and radiologists in the country.

## 2. Materials and Methods

### 2.1. Sampling and Study Population

This research adopts a cross-sectional design conducted at a single centre, employing consecutive sampling to include all patients who underwent multi-detector computed tomography (MDCT) scans for various reasons, including those undergoing medical check-ups without symptoms. Our medical facilities provided MDCT-based medical check-ups to all privately insured patients, which explains the inclusion of some asymptomatic individuals in the screening process. The study spanned from February 2021 to February 2023, with no exclusion criteria applied. Additionally, no control cohort was enlisted, as the primary focus was solely on examining all patients with myocardial bridging (MB).

### 2.2. Protocol for Cardiac MDCT Scanning

Before undergoing scanning, all patients were required to abstain from consuming caffeine and smoking for a minimum of two hours. Additionally, they underwent a renal function assessment and needed approval from radiologists for contrast administration. At our facility, the maximum permissible levels for creatinine and the estimated glomerular filtration rate to facilitate contrast administration were 1.44 mg/dL and 55 mL/min/1.73 m^2^, respectively. Patients with a history of diabetes mellitus who were unable to refrain from metformin intake for eight hours before and 48 h after the procedure would receive consultation from an internist. Due to the iodine content in the contrast, patients also underwent screening for hyperthyroidism (free T3, free T4, and thyroid-stimulating hormone). Lastly, patients and radiologists provided informed consent and disclosed any relevant allergy history.

The subjects’ heart rates ranged from 50 to 100 beats per minute (bpm) on average, measuring 72 ± 13 bpm, irrespective of premedication. Internists or cardiologists administered various types of pre-medication to the patients, such as sublingual nitroglycerin or beta-blockers, two hours before the examination if their heart rate exceeded 100 bpm or if clinically indicated. CCTA was performed using a Siemens Somatom Drive Dual Source—128 slices. Our facility applied specific scanning parameters. An electrography (ECG)-triggered inspiratory CT scan was performed using 100 kV tube voltage and the CARE dose protocol, with the scanned region ranging from the carina to one centimetre below the diaphragmatic face of the heart. For contrast enhancement, our hospital administered 50 mL of Iopamiro (lopamidol 370 mg/mL, Bracco, Milan, Italy), injected at a rate of 5 mL/s via an 18–20 gauge catheter placed in the antecubital vein, followed by a bolus of 50 mL of saline. A bolus contrast test of 10 mL of the same substance and protocol was used when examining the ascending aorta.

Image reconstruction utilised the retrospective ECG-gating method. Data sets were obtained using various percentiles or mS, with the R-R cycle set at 100% or 1000 ms. If the initial results were unsatisfactory, additional window positions within the cardiac cycle were generated. Apart from the axial source images, the image data sets underwent processing on a separate workstation (SyngoVia, Siemens, Forchheim, Germany) and were assessed using thin-slab maximum-intensity-projection (MIP) reconstructions and curved multiplanar reconstruction (MPR) in multiple planes. The coronary artery findings were jointly analysed by a skilled interventional radiologist, one or two thoracic and cardiovascular radiologists, and an experienced cardiology consultant.

### 2.3. Data Collection and Analysis

We characterise a myocardial bridge as the epicardial segment of a coronary artery passing over the heart (see Figure 1A–D). Each segment underwent assessment for the presence of atherosclerotic changes and their location relative to the coronary segment beneath the bridge. The coronary artery calcium score (CAC) was determined using the Agatston method [12]. Stenosis was categorised as follows: <30%, 30–50%, 50–70%, 70–90%, and 100%. In this study, stenosis refers to atherosclerotic plaque lesions visibly obstructing the vessel’s lumen. The location of stenosis was described in relation to the MB segment. All myocardial arteries (including the left anterior descending [LAD], right coronary artery [RCA], posterior descending artery [PDA], left circumflex artery [LCX], and obtuse marginal artery [OM]) were assessed for the presence of myocardial bridge and stenosis (see Appendix A).

### 2.4. Statistical Analysis

The normality of all numerical values would be assessed using the Q-Q plot and Kolmogorov–Smirnov test. If the data followed a normal distribution, mean and standard deviations would be reported, while skewed data would be presented as median and range. Mean values would be compared using an unpaired *t*-test assuming Gaussian distribution with Welch’s correction. Categorical variables would be analysed using the 2 × 2 chi-squared method. Prevalence data would be calculated along with a representative 95% confidence interval derived from the hospital population. GraphPad Prism version 9.0.0 for Windows (GraphPad Software, Boston, MA, USA) was employed for the result analysis.

## 3. Results

The clinical characteristics of our samples can be found in Table 1. There were 2321 cardiac CT scans done in our centre from February 2021 to the end of February 2023, with 1029 patients detected with MB. Hence, the prevalence of MB in our study is 44.3% (95%CI 42.3–46.4). The median age in our cohort is 56 years old (25th quartile: 49, 75th quartile: 63). More males possess MB (62.8% [95%CI 59.7–65.7]) than females (37.2% [95%CI 34.3–40.3]). Symptomatic patients predominate in this cohort, with 67%, and the rest do not have any complaints. The median CAC score is 51 (25th quartile: 0, 75th quartile: 1104.5).

The bridging location occurs overwhelmingly more on the left anterior descending (LAD) vessels, with 99.6%. Within LAD, 4.5% occurs in the distal LAD, and the rest of the bridging happens in mid-LAD. Most patients (59.2%) do not have any stenotic vessels concerning the bridging vessels. Among those with stenosis, the middle portion of the bridging vessel is the most common site of stenosis (*n* = 269), followed by the proximal portion (*n* = 237). The severity of stenosis is more often moderate, with 30–50% (*n* = 238), followed by 50–70% (*n* = 187).

There is a statistically significant difference in the distribution of sex between individuals with stenosis and those without stenosis, as females had a 1.8 times higher risk of having stenosis (95%CI 1.4–2.3; *p*-value < 0.0001). Individuals with stenosis have a significantly higher mean age than those without (*t*-value 5.6, *p*-value < 0.0001). In our study, patients with stenosis have a significantly higher mean CAC score than those without stenosis (*t*-value 11.3, *p*-value < 0.0001). Looking at the reason for scanning, symptomatic patients had a 1.4-times higher risk of having stenosis (95%CI 1.1–1.9; *p*-value = 0.013) (Table 2).

Individuals with more stenotic vessels are more likely to be female than those with fewer stenotic vessels (*t*-value 5.6, *p*-value < 0.0001). Those older and with higher CAC scores are more likely to possess a greater number of stenotic vessels than those with fewer stenotic vessels (*t*-value 185.4, *p*-value < 0.0001 and *t*-value 11.3, *p*-value < 0.0001, respectively). Patients with a greater number of stenotic vessels are also more likely to be scanned for symptomatic reasons compared to those with fewer stenotic vessels (*t*-value 5.6, *p*-value < 0.0001) (Table 3). The degree of stenosis is significantly higher in the proximal stenosis group than in the middle stenosis group (*t*-value 27, *p*-value < 0.0001).

## 4. Discussion

There is a moderately high prevalence of MB in our centre, with a male predominance and in the LAD location. The middle portion of the bridging vessel is the most commonly implicated site of stenosis, with most patients having a moderate (30–50%) degree of stenosis. Females, people of older age, those with higher mean CAC scores, and symptomatic patients are significantly more likely to have stenosis.

The prevalence in this study is higher than reported in one meta-analysis, where they reported a pooled prevalence rate of 32.9% [3]. Our prevalence is also higher than the only other study in Indonesia, where they found a 16.3% prevalence of MB amongst 934 patients [10]. There is a lack of awareness of MB in Indonesia, and it is thought to be a rare finding with unknown clinical consequences [11]. Males have an overwhelmingly higher prevalence in our study, and this finding is supported by the other study conducted in Indonesia [3,10]. However, other studies suggest that the prevalence is higher in females [13] or relatively similar among males and females [14]. However, females in our study are statistically more likely to have stenosis and have a higher number of stenotic lesions. The reason for this is unknown, and the results of our study could not explain this. Other studies looked at MB and non-MB patients, and the comparison among MB patients may be why this finding is the first in the literature.

The gold standard in the diagnosis of MB is autopsy studies [4,15] while in vivo high-resolution CT scans are preferred over coronary angiography studies [4]. Coronary angiography tends to underestimate the prevalence of MB because investigators must depend on indirect indicators to assess the vessel. While systolic compression, the milking effect, and the step-down–step-up phenomenon serve as diagnostic signs, they exhibit limited sensitivity in shallow variants of MB [16]. Furthermore, Indonesia is a country characterised by substantial limitations in infrastructure, human capital, and financial resources in terms of conducting advanced cardiovascular research. Hence, coronary angiography is seldom used for research or treatment-related purposes [17]. Therefore, the use of MDCT as a method of diagnosing MB can be very promising in Indonesia.

The significance of MB is still up in the air for debate. Since MB is the most common congenital coronary anomaly, it is bound to be found in many patients [5]. Hence, many researchers are linking MB with a variety of adverse cardiac events, such as myocardial infarction, life-threatening arrhythmias, hypertrophic cardiomyopathy [18], and sudden cardiac death [3,8,9]. However, some researchers also believe that MB may be a protective factor [3], as one study by Loukas [19] found weak inflammation activity in the pre- and post-myocardial bridge segments and decreased numbers in myocardial bridge segments. 

The LAD is the most commonly involved coronary artery in this study, already known in the literature [20]. However, the prevalence is much higher than the reported literature, with 5% to 86%, in contrast to 99.6% in our study [21]. One explanation for this difference is the modality used in diagnosing LAD-MB, where our study used MDCT and the other study used necropsy to diagnose MB [21]. One study that also used MDCT found that 91% (*n* = 78) of MBs occur in the LAD [14]. 

The CAC score is used in this study as a proxy marker for atherosclerosis, and the median result shows that patients have a low risk of cardiovascular events [22]. One study found that more patients without bridging (11%) have a higher calcium score of >400 than patients without bridging (3%), and the difference is statistically significant [10]. One explanation for their finding is that their study had five times more patients without MBs than with MBs. Another explanation is that they recruited all patients with suspected or known coronary artery disease, which explained the higher number of patients with high CAC scores. In our study, a third of our samples are asymptomatic, and the median age is relatively young, which may explain the low CAC score. 

Our results show that higher mean CAC scores are found in patients with stenosis, and patients with more stenotic vessels have a higher mean CAC score. This finding can be explained by endothelial dysfunction. One of the initial stages of atherogenesis is endothelial dysfunction [23]. Many investigations have documented modifications to the typical endothelial physiology linked to MB, such as an atypical reaction to acetylcholine (ACh) or reduced responses to endothelin-1 and nitric oxide. If the endothelium is intact, acetylcholine releases endothelial-derived relaxation factor, acting as an endothelium-dependent vasodilator [24]. However, ACh will cause smooth-muscle constriction if endothelial lesions are present, including atherosclerotic disease [25]. However, this finding differs from another study conducted in Indonesia, where patients with MB have fewer coronary stenoses [10]. This may be attributed to how they sample their patients. 

Atherosclerosis has been linked to the presence of MB, and the mechanism underlying this relationship is likely related to shear stress. Reducing shear stress can promote endothelial dysfunction and atherosclerosis [26]; systolic flow interruption modifies laminarity by reducing shear stress upstream of the tunnelled section; conversely, increased flow velocities cause more significant shear stress on the intramyocardial segment. Consequently, the intra-myocardial segment would be preserved, but the tract just upstream of it could be more susceptible to atherosclerosis if MB is present [27]. This theory supports our study finding that the degree of stenosis is significantly higher in the proximal stenosis group than the middle stenosis group. One study involving 25 patients found that, among individuals with MB, inflammation is elevated in proximal epicardial adipose tissue (EAT) when compared to bridge EAT, but this disparity is observed only among those with a high burden of atherosclerotic plaque. Inflammation was evidenced through both physical traits (computed tomography attenuation of the pericoronary fat) and molecular features (mRNA expression and secretion of cytokines) [28].

One of the limitations of this study is the lack of data on the length and maximum myocardial thickness overlying the coronary artery at the site of the bridge. There are also no data on the diameter of each tunnelled segment in both the end-diastolic and end-systolic phases. The degree of systolic compression and arterial route is related to, but not the only predictor of, the depth of the tunnelled artery (1–2 mm is considered superficial and >2 mm is considered deep). Treatment decisions are also impacted by the MB’s depth, particularly when a surgical procedure is being considered. The length of the tunnelled segment is significant since it affects both the number of branches impacted by the MB and the volume of the affected artery. This is particularly pertinent from a therapeutic standpoint when analyzing LAD MBs that impact septal or diagonal branches [15]. The lack of this data may serve as a confounder that affects our results.

There are several limitations in our studies. First, we could not ascertain causality since this is a cross-sectional study. The nature of our study also means that we have no follow-up data on the morbidity or mortality of our patients. Second, due to the retrospective nature of data collection, some data could not be collected, such as the presence of comorbidities, such as diabetes mellitus, hypercholesterolemia, or smoking status. However, other studies did not find any meaningful associations between these risk factors and MB [10,14]. Therefore, the absence of these risk factors may not be meaningful in our study. Third, our studies did not further confirm the presence of MB with other modalities, such as using optical coherence tomography or intravascular ultrasound [15]. The gold standard to diagnose MB is through autopsy, but this method could not be used in our study [29]. Fourth, as MB is still not a widely known finding in Indonesia, the standard reporting for MB is not yet universal. This fact is evident in the previous study in Indonesia, which also lacked such reporting and data [10]. Through this study, we hope a more standardised reporting on myocardial bridging could be developed in Indonesia. The last limitation is related to the use of MDCT. Since each 3D volume set’s data is gathered over several heartbeats, this technique’s usefulness is reliant on a steady heart rate and negatively impacted by arrhythmias, such as numerous ectopic beats and atrial fibrillation. Some of our patients presented with arrhythmia. This may introduce some false positives or false negatives, as they affect the image quality. However, the use of arrhythmia adjustment software and pre-medication should have mitigated this issue.

## 5. Conclusions

Our study has a high prevalence (44.3%) of myocardial bridging (MB). Additionally, our research demonstrates that MB may provide protection against atherosclerosis of the coronary distal segment to the MB and trigger the development of atherosclerosis in the segment proximal to the bridge. We also encourage a more standardised reporting on MB, as the assessment of the depth, length, and thickness of MB is crucial to advancing research in this area.

## Figures and Tables

**Figure 1 medicina-60-00794-f001:**
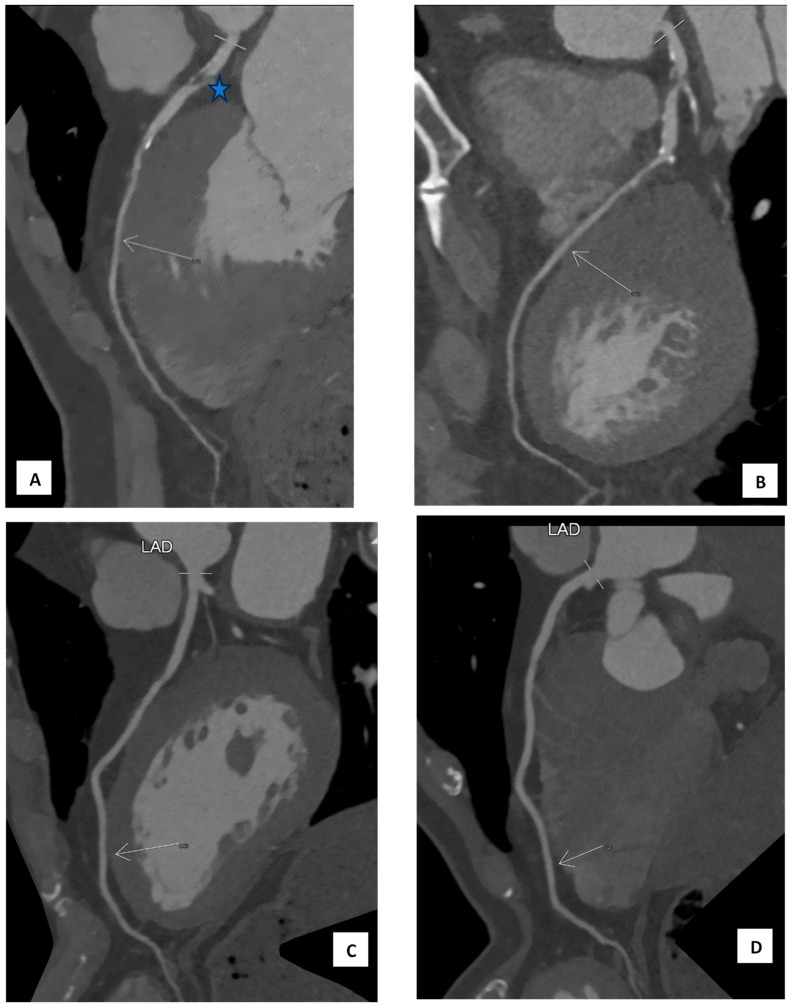
(**A**,**B**) The MDCT of a 59-year-old male with chest discomfort shows a band of myocardial muscle overlying the mid-left anterior descending artery (LAD) segment corresponding to a myocardial bridge (arrows). There is a 70–90% stenosis proximal to the bridging (star). The calcium artery score (CAC) is 312.9. (**C**,**D**) show the MDCT of a 60-year-old male without any symptoms (medical check-up), which shows a superficial band of myocardial muscle overlying the mid-LAD segment corresponding to the myocardial bridge (arrows). There is no stenosis, and the CAC score is 0.

**Table 1 medicina-60-00794-t001:** Descriptive characteristics of the myocardial bridging patients.

Variables	Number (%)
Median age (years)	56 (25th quartile: 49, 75th quartile: 63)
Sex	
Male	646 (62.8)
Female	383 (37.2)
Median coronary artery calcium score	51 (25th quartile: 0, 75th quartile: 1104.5)
Reason for scanning	
Symptomatic	689 (67)
Asymptomatic	340 (33)
Vessel bridging location	
Mid LAD	979 (95.1)
Distal LAD	46 (4.5)
Distal LCX	1 (0.1)
Mid RCA	1 (0.1)
Obtuse marginal	2 (0.2)
Number of stenotic vessels	
0	609 (59.2)
1	307 (29.8)
≥2	113 (11)
Location of stenotic vessels (*n* = 420) *	
Proximal	237
Middle	269
Proximal-middle	8
Distal	4
Severity of stenotic vessels (%)	
<30	18/420
30–50	238/420
<50	2/420
50–70	187/420
70–90	88/420
100	2/420

* One patient can have more than one location for the stenotic vessels; LAD, left anterior descending; LCX, left circumflex, RCA, right coronary artery.

**Table 2 medicina-60-00794-t002:** Analytical results in patients with myocardial bridging with the presence of stenosis being the dependent variable.

Independent Variable	*t*-Value	Odds Ratio (95% CI)	*p*-Value
Sex	-	1.8 (1.4–2.3)	<0.0001
Age	5.6	-	<0.0001
Coronary artery calcium score	11.3	-	<0.0001
Reason for scanning	-	1.4 (1.1–1.9)	0.013

CI: confidence interval.

**Table 3 medicina-60-00794-t003:** Analytical results in patients with myocardial bridging with the number of stenotic vessels being the dependent variable.

Independent Variable	*t*-Value	*p*-Value
Sex	5.6	<0.0001
Age	185.4	<0.0001
Coronary artery calcium score	11.3	<0.0001
Reason for scanning	5.8	<0.0001

## Data Availability

The data will be made available upon reasonable request.

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
