# Peer review of "High Prevalence of Myocardial Bridging Detected in an Indonesian Population Using Multi-Detector Computed Tomography"

_medicina, 2024, doi:10.3390/medicina60050794_

Round 1
Reviewer 1 Report (New Reviewer)
Comments and Suggestions for Authors
Myocardial bridging is a common anatomical variant in which an epicardial coronary artery takes an intramyocardial course, leading to dynamic systolic compression. Because coronary perfusion occurs primarily during diastole, most patients with this anatomical variant have no associated perfusion abnormalities or symptoms. Despite this, there is a subset of patients with myocardial bridging who experience myocardial ischaemic symptoms. Determining which anatomical variants are benign and which are clinically relevant remains a challenge. Further complicating the picture, functional factors such as diastolic dysfunction and coronary vasospasm may exacerbate myocardial bridging-related ischemia. In patients with ischaemic symptoms in the absence of alternative explanations, a detailed assessment of myocardial bridging with invasive physiology should be performed to define the significance of the lesion and guide tailored medical therapy [Danek BA, Kearney K, Steinberg ZL. Clinically significant myocardial bridging. Heart. 2023 Dec 20;110(2):81-86. doi: 10.1136/heartjnl-2022-321586.].
The prevalence of myocardial bridging is a matter of debate as reports vary due to differences in diagnostic technique. By autopsy, the pooled prevalence of myocardial bridging was found to be 42% [Hostiuc S, Negoi I, Rusu MC, Hostiuc M. Myocardial Bridging: A Meta-Analysis of Prevalence. J Forensic Sci. 2018 Jul;63(4):1176-1185. doi: 10.1111/1556-4029.13665.], although variation in definition led to heterogeneity of estimates. The pooled prevalence of myocardial bridging by invasive coronary angiography is much lower, estimated at 6%. This highlights variability in detection by angiography depending on factors that influence the degree of vessel compression, including length and depth of the tunneled segment, orientation of muscle fibers with respect to the course of the vessel and patient hemodynamics. Computed tomography demonstrated less heterogeneity between reports than autopsy or invasive angiography, with a pooled prevalence estimate of 22%, likely due to more uniform definitions of what constitutes an intramyocardial bridge [Danek BA, Kearney K, Steinberg ZL. Clinically significant myocardial bridging. Heart. 2023 Dec 20;110(2):81-86. doi: 10.1136/heartjnl-2022-321586.].
The aim of the present article to assess the prevalence of myocardial bridging (MB) using multi-detector computed tomography (MDCT) and look at factors contributing to stenosis amongst patients with MB. According to study results authors showed high prevalence (44.3%) of MB. Additionally, their research demonstrates that MB may provide protection against atherosclerosis of the coronary distal segment to the MB and trigger the development of atherosclerosis in the segment proximal to the bridge.
The manuscript is clear, relevant for the field and presented in a well-structured manner.
The 29 (100%) references are given. The 14 (48.3%) cited references is not mostly recent publications (within the last 5 years). An optional recommendation is to replace some outdated references with newer ones, preferably no more than 5 years old. Article is not including an excessive number of self-citations.
The manuscript is scientifically sound and the experimental design is appropriate to test the hypothesis.
The manuscript’s results are reproducible based on the details given in the methods section.
The figures and table are appropriate. They are properly showed the data. They are easy to interpret and understand. The data is interpreted appropriately and consistently throughout the manuscript.
The conclusions are consistent with the evidence and arguments presented.
The ethics statements and data availability statements are adequate.
Author Response
Please see the attachment.

Reviewer 2 Report (New Reviewer)
Comments and Suggestions for Authors
I thank the editor for having the opportunity to review an extremely interesting manuscript describing MB in general population in Indonesia.
The text of the present manuscript is well organized, written in accordance to English syntax and with sufficient overview of the existing scientific literature.
· In describing technical specification of imaging performed in all patients, I have noticed that the tube current time was set at 247 mAs. Generally, the mAs provided by the CT scanner if operating in the CARE regime do change on a millisecond scale and are tissue dependant. I would kindly ask the authors to state this more clearly or to re-write the paragraph on technicalities and simply write something like: “ECG triggered inspiratory CT scan was performed using 100 kV tube voltage and CARE dose protocol, with the scanned region ranging from the carina to one centimetre below the diaphragmatic face of the heart.”
Author Response
Please see the attachment

This manuscript is a resubmission of an earlier submission. The following is a list of the peer review reports and author responses from that submission.
Round 1
Reviewer 1 Report
Comments and Suggestions for Authors
1. Keywords of "Coronary imaging" is not a MeSH term. Please correct it to "Cardiac-Gated Imaging Techniques".
2. Another important reference for the first sentence should be considered: JACC: Myocardial Bridging: Diagnosis, Functional Assessment, and Management: JACC State-of-the-Art Review.
3. Rephrase "The exact prevalence of this condition depends of this condition depends on........"
4. The description of the prevalence of MB is better described according to escalating percentage, i.e.: 4.3% then 21.7%, and then 44.6%. The reference of prevalence should be considered: J Forensic Sci: Myocardial Bridging: A Meta-Analysis of Prevalence.
5. Line 57: what is CDCT?
6. Line 73: creatine? or creatinine?
7. Line 74: ")" ??
8. Statistical analysis: please describe the analysis method for categorical parameters.
9. Line 127: How was the 95% CI calculated? What was your population parameter? The whole national population? Whole hospital population? or?
10. Please explain for the first appearance of "CI"
11. Result: Why do you use t-test for categorical parameter?
12. Discussion Line 178: The reference studying inflammation and MB should be considered: JAHA: Relationship Between Coronary Atheroma, Epicardial Adipose Tissue Inflammation, and Adipocyte Differentiation Across the Human Myocardial Bridge.
13. Please describe the depth (e.g.: mild<2mm, moderate 2-5mm, severe >5mm) and length of tunneled vessels
14. The protocol of MDCT was somewhat over-detailed and redundant. The anatomical description of MB was inadequate.
15. The figures could be more brief and concise.
Comments on the Quality of English LanguageThere are some grammar misses. The description sentences could be better. Please send professional English editing.
Author Response
Author’s Response:
We thank the reviewers for their time and expertise. Please see the specific response to the reviewers enclosed below.
Reviewers' Comments to Authors:
1. Keywords of "Coronary imaging" is not a MeSH term. Please correct it to "Cardiac-Gated Imaging Techniques".
Response: Thank you for your insight. We have changed the keyword accordingly.
- Another important reference for the first sentence should be considered: JACC: Myocardial Bridging: Diagnosis, Functional Assessment, and Management: JACC State-of-the-Art Review.
Response: Dear reviewer, this reference has already been cited in our manuscript under the number of reference number 24. Thank you.
- Rephrase "The exact prevalence of this condition depends of this condition depends on........"
Response: Thank you for your input. We have rephrased the following sentence into:
“The precise occurrence of this condition varies according to the diagnostic method employed for MB, with rates of 4.3% with conventional angiography, 21.7% using coronary computed tomography angiography (CCTA), and 44.6% with autopsy or cadaveric dissection”
- The description of the prevalence of MB is better described according to escalating percentage, i.e.: 4.3% then 21.7%, and then 44.6%. The reference of prevalence should be considered: J Forensic Sci: Myocardial Bridging: A Meta-Analysis of Prevalence.
Response: Thank you for your input. We have rearranged the number as indicated in revision point number 3 and we have also added a new sentence with the reference as suggested by the reviewer:
“Another meta-analysis of 120 studies found that the total prevalence stood at 19% (confidence interval [CI]: 17–21%); autopsy investigations indicated a total prevalence of 42% (CI: 30–55%), CT studies showed 22% (CI: 18–25%), and coronary angiography indicated a prevalence of 6% (CI: 5–8%)”
- Line 57: what is CDCT?
Response: Thank you for your astute observation. It was indeed an error on our part and we have changed CDCT to MDCT.
- Line 73: creatine? or creatinine?
Response: Thank you for your input. We have changed creatine into creatinine.
- Line 74: ")" ??
Response: Thank you for your astute observation. We have deleted the closed brackets
- Statistical analysis: please describe the analysis method for categorical parameters and 11. Result: Why do you use t-test for categorical parameter?
Response: Thank you for your input and we are sorry for this oversight. We have added how we analyze categorical variable as follows
“The categorical variables are calculated using the 2x2 Chi Squared method.”
We also have updated our results and table 2 accordingly. As for the part of “dependent variable : number of vessels”, we follow the following paper as the reasoning of using t-test: “Zeina et al (2006) Myocardial Bridge: Evaluation on MDCT”
- Line 127: How was the 95% CI calculated? What was your population parameter? The whole national population? Whole hospital population? or?
Response: Thank you for your comment. We have added a sentence to make it clearer as follows:
“The prevalence data would be calculated with a representative 95%CI derived from the hospital population”
- Please explain for the first appearance of "CI"
Response: Thank you for your input. As per input for number 4, we have explained the first appearance of CI there.
- Discussion Line 178: The reference studying inflammation and MB should be considered: JAHA: Relationship Between Coronary Atheroma, Epicardial Adipose Tissue Inflammation, and Adipocyte Differentiation Across the Human Myocardial Bridge.
Response: Thank you for the input. We have read the literature recommended and have added the following sentence
“One study involving 25 patients found that among individuals with MB, inflammation is elevated in proximal epicardial adipose tissue (EAT) when compared to bridge EAT, but this disparity is observed only among those with a high burden of atherosclerotic plaque. Inflammation was evidenced through both physical traits (computed tomography attenuation of the pericoronary fat) and molecular features (mRNA expression and secretion of cytokines)”
- Please describe the depth (e.g.: mild<2mm, moderate 2-5mm, severe >5mm) and length of tunneled vessels
Response: Thank you for your valuable insight. However, we could not do this at the current stage of research and it has been explained in our limitation section. We have acknowledged this in our first version of the manuscript as follows:
“One of the limitations in this study is the lack of data on the length and maximum myocardial thickness overlying the coronary artery at the site of the bridge”
And also we have emphasized this in the conclusion section
“We also encourage a more standardized reporting on MB, as the assessment of the depth, length, and thickness of MB is crucial to advancing research in this area.”
- The protocol of MDCT was somewhat over-detailed and redundant. The anatomical description of MB was inadequate.
Response: Thank you for your comment. We followed the protocol of MDCT according to this literature ““Zeina et al (2006) Myocardial Bridge: Evaluation on MDCT”” and we believe a strong protocol on MDCT is the key to reproducibility hence we would like to keep the description. Besides, the Associate Editor commented on our manuscript being “too short” and hence reducing the protocol would mean decreasing the count number even more. As for the anatomical description of MB, we have admitted again in our limitation that this is lacking and we also emphasized it again in the conclusion section.
- The figures could be more brief and concise.
Response: Thank you for your comment. As we mentioned in point number 14, making the figures more brief and concise would limit the readers in understanding of our manuscript (as the figures contain all the anatomical details of the coronary arteries) and may potentially reduce the reproducibility. Hence, we prefer to keep the figures as it is.
We have improved our English language (double-checked by an English native as well as a researcher with five years of experience with an Editorial background – GSO), with the assistance of Grammarly. We hope the language is good enough and we will be more than happy to revise specific sentences that are not good enough for the reviewer. Thank you!

Reviewer 2 Report
Comments and Suggestions for Authors
This article promoted the particular topic and should also be addressed why this method was important to highlight comparing with access and results of coronary angiography.
In addition, it was not clear and I suggest adding the sentence about "Patients' clinical characteristics were presented in table 1".
I recommend authors to expand "materials and methods" part by adding how patients with no clinical symptoms were chosen to the study, did they receive any heart treatment and what were exclusion criteria. Benefits and limitations between 2 methods (CAG and CT) should be compared.
I also suggest adding an algorithm for how many patients were scanned, then included and other following actions.
Comments on the Quality of English LanguageI recommend checking this manuscript by native English speaker.
Author Response
Author’s Response:
We thank the reviewers for their time and expertise. Please see the specific response to the reviewers enclosed below.
Reviewers' Comments to Authors:
This article promoted the particular topic and should also be addressed why this method was important to highlight comparing with access and results of coronary angiography.
Response: Thank you for your insight. We have added the following paragraph in response to this input:
“The gold standard in the diagnosis of MB is autopsy studies(4, 15) while in-vivo high-resolution CT scans are preferred over coronary angiography studies.(4) Indonesia is a country characterized by substantial limitation in infrastructure, human capital, and financial resources in terms of conducting advanced cardiovascular research. Hence, coronary angiography is seldom used for research or treatment-related purposes.(16) Therefore, the use of MDCT as a method of diagnosing MB can be very promising in Indonesia.”
In addition, it was not clear and I suggest adding the sentence about "Patients' clinical characteristics were presented in table 1".
Response: Thank you for your input. We have inserted the following sentence:
“The clinical characteristics of our samples could be found in Table 1”
I recommend authors to expand "materials and methods" part by adding how patients with no clinical symptoms were chosen to the study, did they receive any heart treatment and what were exclusion criteria. Benefits and limitations between 2 methods (CAG and CT) should be compared.
Response: Thank you for your input. We have added the following sentence regarding on why asymptomatic patients were scanned
Our hospitals offered a medical check-up using MDCT to all private and insured patients, which explain why some patients who claimed to be asymptomatic were screened
As for the exclusion criteria, we have explicitly said in our manuscript that “This study has no exclusion criteria, and we did not recruit a control cohort as this study aims to look at all patients with MB only”
We have added the comparison on CAG with the following sentence:
Coronary angiography tends to underestimate the prevalence of MB because investigators must depend on indirect indicators to assess the vessel. While systolic compression, the milking effect, and the step down–step up phenomenon serve as diagnostic signs, they exhibit limited sensitivity in shallow variants of MB
I also suggest adding an algorithm for how many patients were scanned, then included and other following actions.
Response: Thank you for your input. As noted in our manuscript, we have included this following sentence “There are 2,321 cardiac CT scans done in our centre from February 2021 to the end of February 2023, with 1,029 patients detected with MB”. Since we did not have any exclusion criteria, hence there are no following actions. We have considered adding a CONSORT diagram before but since it would be just a simple 2,321 cardiac scans leading to 1,029 patients being diagnosed with myocardial bridging, we decided not to include the algorithm.
We have improved our English language (double-checked by an English native as well as a researcher with five years of experience with an Editorial background – GSO), with the assistance of Grammarly. We hope the language is good enough and we will be more than happy to revise specific sentences that are not good enough for the reviewer. Thank you!

Round 2
Reviewer 1 Report
Comments and Suggestions for Authors
1. A professional English editing service was strongly recommended.
2. Why do readers have to view so many similar pictures of cardiac vessels on CT? Figure 2(A) was like 2(B). The "OM1" you showed in Figure 2(I) was actually an intermediate artery. Are these large and redundant pictures necessary in an article discussing MB?
3. You have the images of each patient. Why can't you measure the depth and length of MB?
4. Since you searched for risk factors for "stenotic vessels" (I think you mean atherosclerotic coronary artery disease), why not collect clinical information on T2DM, HTN, dyslipidemia, and smoking status?
5. Why do readers have to know t-values? Do any other articles show t-values?
6. The table presentation was also not in a scientific form.
7. What do you mean by "stenosis"? atherosclerotic coronary artery disease? coronary hypoplasia? myocardial bridging compression narrowing?
Comments on the Quality of English Language
Please search for a professional English editing service.
Author Response
Please look at our response to reviews

Reviewer 2 Report
Comments and Suggestions for Authors
Revised manuscript significantly improved.
Author Response
Thank you for your kind input